# A multi-domain collaborative denoising bearing fault diagnosis model based on dynamic inter-domain attention mechanism and noise-aware loss function

**Weilin Cao** [ID]*, **Liqiang Zhang**

School of Artificial Intelligence, Neijiang Normal University, Sichuan, China

* caoweilin85@163.com

## Abstract

Rolling bearings are the core transmission components of large-scale rotating machinery such as wind power gearboxes and aviation engines, so timely and effective monitoring and diagnosis of their status are crucial to ensure the stable operation of equipment, reduce maintenance costs, and improve production efficiency. However, the noise interference in the industrial field often hides the original characteristics of the bearing fault signal, leading to the deep learning-based fault diagnosis model's lack of diagnostic reliability in the strong industrial noise background. To address this problem, this paper proposes a multi-domain collaborative denoising diagnostic model based on dynamic inter-domain attention mechanism and noise-aware loss function. First, the model extracts high-dimensional features of bearing fault signals from multiple domains, such as time and frequency domains, aiming to enhance the richness and diversity of high-dimensional features to effectively suppress noise interference on the diagnostic results. Second, the dynamic inter-domain attention mechanism (DIDAM) is proposed, aiming to distinguish the importance of information in different signal domains and flexibly integrate them to realize more efficient and accurate multi-domain information fusion. Finally, the noise-aware loss function (NALF) is designed to avoid the phenomenon of the conduction model being prone to making wrong decisions due to excessive noise. Experimental results on two publicly available datasets, CWRU and MFPT, show that even in the extreme noise environment with SNR = −10 dB, the proposed model still achieves 81.25% and 76.36% fault diagnosis accuracies, which are better than most existing mainstream denoising models. Overall, the proposed method can still perform well under substantial noise interference, providing a new idea for intelligent bearing fault diagnosis in real industrial scenarios.

## Introduction

Rolling bearings are widely used to support rotating shafts and their components due to their strong load-bearing capacity, high motion accuracy, and low friction loss, and therefore

**Data availability statement:** The CWRU and MFPT datasets used in this study are available on Figshare: the CWRU dataset can be

downloaded from
10.6084/m9.figshare.28606778, and the MFPT
dataset can be downloaded from
10.6084/m9.figshare.28606802.

**Funding:** This research was supported by the
Research and Innovation Team Project of
Neijiang Normal University (No. 18TD02). We
confirm that the funders had no role in study
design, data collection and analysis, decision to
publish, or preparation of the manuscript.

**Competing interests:** The authors have
declared that no competing interests exist.

are known as the "joints" of rotating machinery and equipment [1]. However, the working
environment of rolling bearings is usually more complicated, making them very prone to fail-
ure, which may lead to equipment downtime and even cause serious safety accidents [2–4].
In conclusion, the condition of rolling bearings directly affects the operating efficiency,
reliability, and safety of mechanical systems, so timely and effective monitoring and diagnosis
are essential to ensure the regular operation of equipment [5,6].

Signal processing and machine learning are two common strategies in bearing fault diag-
nosis. For instance, Wang et al. [7] proposed a fault diagnosis method based on feature vector
migration learning. This method involves feature extraction from vibration signals through
statistics and wavelet decomposition, then using the ReliefF algorithm to evaluate and select
sensitive fault feature sets. Salunkhe et al. [8] pointed out that the dynamic characteristics of
rolling bearings are mainly influenced by their geometric structure and operating parameters,
and they developed a bearing fault diagnosis method based on the Hilbert-Huang transform.
Thamba et al. [9] developed a new method for analyzing the fault patterns of self-aligning
bearings and achieved automatic fault classification through artificial neural networks and
deep neural networks. Goyal et al. [10] denoised the signals using the Hilbert transform, then
applied principal component analysis for dimensionality reduction, followed by the sequen-
tial floating forward selection to screen the optimal feature set, and finally, these features
were input into support vector machines and artificial neural networks for fault recognition
tasks.

Signal processing-based methods often rely on prior understanding or assumptions about
signal features and may not effectively extract relevant features when dealing with nonlinear
and non-stationary signals, thus affecting the accuracy of fault diagnosis. Although machine
learning-based methods can automatically extract features, manual feature engineering is
still required in many cases, which demands professional knowledge, is time-consuming, and
inappropriate feature selection may affect the model's performance.

In recent years, with the continuous upgrade of computer hardware and the increasing
maturity of cutting-edge theories such as artificial intelligence, deep learning has become
the dominant force in bearing diagnosis research due to its end-to-end training model and
outstanding feature representation capabilities [11]. Huang et al. [12] pointed out that tradi-
tional Convolutional Neural Networks (CNNs) are prone to falling into local optima when the
input signals lack practical information. To address this issue, they proposed the Multi-scale
Cascaded Convolutional Neural Network (MC-CNN), which enhances the diversity of sig-
nal representation in the frequency domain by introducing filters of different scales, thereby
providing more valuable information for fault diagnosis. He et al. [13] believe that the fixed
convolution kernel parameters in traditional CNNs limit their ability to extract key features
from fault signals. Therefore, they introduced IMSCNN, which uses dilated convolution
kernels with different dilation rates to extract multi-scale features, thereby improving the
effectiveness of bearing fault diagnosis. Song et al. [14] proposed the strategy of widening
convolution kernels to expand the receptive field and designed the WKCNN network struc-
ture based on this idea. Experimental results show that WKCNN outperforms other methods
regarding diagnostic accuracy and timeliness. Hou et al. [15] proposed Diagnosisformer, an
attention-based multi-feature parallel fusion model for rolling bearing fault diagnosis. This
method extracts frequency domain features through the Fast Fourier Transform (FFT) and
designs a multi-feature fusion module to capture more information from different recep-
tive fields, extract essential features, and maintain global dependency relationships. Guo et
al. [16] proposed an end-to-end fault diagnosis method that combines Attention CNN with
a Bidirectional Long Short-Term Memory network (BiLSTM) (ACNN-BiLSTM). This method
extracts short-term spatiotemporal features through one-dimensional wide convolution and

introduces a convolution block attention module to adjust the weights of different feature dimensions dynamically, finally inputting the weighted features into BiLSTM for fault diagnosis. Liao et al. [17] proposed the MSRN-EGRU model, a hybrid deep learning model that combines a Multi-scale Residual Neural Network (MSRN) with an Enhanced Gated Recurrent Unit (EGRU). Experimental results show the model's accuracy, robustness, and convergence in bearing fault diagnosis.

However, the studies above did not consider the noisy and harsh working conditions bearings face in industrial environments. In industrial applications, bearings often operate against a backdrop of strong noise, such as high-speed operation of equipment, external environmental noise, and vibrations from other mechanical components [18–20]. These noise interferences can mask the original characteristics of the bearing fault signal, making it difficult to accurately identify fault types, significantly affecting the accuracy and robustness of deep learning-based fault diagnosis methods [21,22].

Researchers have proposed a series of anti-noise models to address noise interference in bearing fault diagnosis. Li et al. [23] have proposed a clustering algorithm based on deep convolutional neural networks, which reduces the discrepancy between training and testing data by minimizing intra-cluster variance and maximizing inter-cluster variance, thereby enhancing the model's robustness under noise and varying working conditions. Chen et al. [24] introduced the MCNN-LSTM model, which uses multi-scale convolution kernels to extract high-frequency and low-frequency components of vibration signals and identifies the type of fault through a Long Short-Term Memory network (LSTM). Hakim et al. [25] transformed the signal from the time domain to the frequency domain using Fast Fourier Transform (FFT), separated the amplitude and phase using phase representation, and then inputted it into a 1D-CNN to enhance the model's anti-noise capability. Li et al. [26] proposed an end-to-end Adaptive Multi-scale Full Convolutional Network (AMFCN), which improves adaptability to noise through random sampling and employs large convolution kernels for wide-range temporal feature extraction, effectively suppressing high-frequency noise and significantly improving the model's anti-noise performance and robustness, outperforming traditional CNNs and other multi-scale CNN models. Wang et al. [27] designed an Adaptive Denoising Convolutional Neural Network (ADCNN), which removes noise through adaptive denoising units while preserving key fault features, and further enhances anti-noise capability by reducing the number of channels and increasing the size of convolution kernels. Han et al. [28] proposed a Deep Residual Multi-scale Convolutional Neural Network with an attention mechanism (Attention-MSCNN), which effectively removes noise through residual connections and a denoising multi-head attention mechanism and captures the relationships in long-term sequences. Gao et al. [29] proposed an Adaptive Global-Local Denoising Multi-Time Scale Attention Residual Shrinkage Network (AMARSN), which enhances discriminability under noise through a multi-time scale attention learning module and removes noise from multi-scale fault features through an adaptive denoising module.

Although existing denoising models have made some progress, they still have many shortcomings when faced with strong noise environments. Firstly, most methods focus solely on processing time-domain signals, neglecting key information contained in other potential domains. This results in the model's difficulty in accurately extracting the original features of fault signals under strong background noise, limiting its ability to express high-dimensional features of different fault patterns. Secondly, although some methods attempt to extract signal features from different domains, they rely solely on fixed feature concatenation or fusion approaches, failing to fully address the differential contributions of features from different signal domains to model performance, leading to poor multi-source information fusion. Lastly, existing denoising models typically use the standard cross-entropy function

for network parameter updates, neglecting the possibility that strong noise environments may lead the model to make incorrect decisions, resulting in deviations in the direction of gradient updates. Therefore, this paper proposes a multi-domain collaborative denoising bearing fault diagnosis model based on a dynamic inter-domain attention mechanism and a noise-aware loss function, with the following contributions:

1. A multi-domain collaborative denoising bearing fault diagnosis model is proposed, which can simultaneously extract high-dimensional bearing fault signal features from different domains. The model enhances the richness and diversity of high-dimensional feature expression in the original signals, effectively improving the model's fault recognition ability under substantial noise interference.
2. Given the expressive differences and complementary relationships among multi-domain features, a dynamic inter-domain attention mechanism is designed to achieve more efficient information fusion. This mechanism can flexibly adjust the fusion approach according to the importance of information in each signal domain, improving the efficiency and accuracy of information fusion.
3. Considering that excessive noise may lead the model to make incorrect decisions, resulting in deviations in the direction of gradient updates, a noise-aware loss function is constructed. This loss function can avoid deviations in the model's gradient update direction caused by strong noise environments, thereby enhancing the stability and reliability of fault diagnosis.

## Fundamental theories

This section elaborates on the three fundamental theories that underpin this research in terms of principles and functionality: the Fast Fourier Transform, the Squeeze-and-Excitation Attention Mechanism, and the Cross-Entropy Loss Function. These theories provide crucial theoretical support and inspiration for the model proposed in this paper and facilitate a deeper understanding of its design and implementation.

### Fast Fourier transform

In the field of fault diagnosis for rotating machinery, time-domain vibration signals often exhibit nonlinear and non-stationary characteristics, and their transient impact components are easily disrupted by background noise. Traditional time-domain analysis methods are sensitive to random noise, making extracting periodic fault features stably challenging.

As a core tool in signal processing, the Discrete Fourier Transform (DFT) [30] can convert a discrete signal from the time domain to the frequency domain, thereby extracting frequency domain features and providing mathematical support for fault diagnosis in noisy environments. Specifically, for a discrete signal $x(n)$ of length $N$, its DFT is defined as follows:

$$X(k) = \sum_{n=0}^{N-1} x(n)e^{-j\frac{2\pi}{N}kn}, \quad k = 0, 1, \ldots, N-1 \tag{1}$$

Here, $X(k)$ represents the spectral value of the signal at the k-th discrete frequency point, and $e^{-j\frac{2\pi}{N}kn}$ is the rotation factor (an element of the rotation matrix), which can also be expressed as:

$$W_N^{kn} = e^{-j\frac{2\pi}{N}kn} \tag{2}$$

However, directly computing the DFT requires $N^2$ complex multiplications and additions, which leads to a substantial computational burden for large $N$. The Fast Fourier Transform (FFT) [31] is an efficient algorithm for computing the DFT. It employs a divide-and-conquer strategy, drastically reducing the computational cost from $O(N^2)$ to $O(N log N)$ and improving efficiency.

Specifically, the FFT decomposes a DFT of length $N$ into two DFTs of length $N/2$, thereby splitting the summation of a DFT of length $N$ into odd and even parts, as follows:

$$X(k) = \sum_{n=0}^{N/2-1} x(2n) W_N^{2kn} + \sum_{n=0}^{N/2-1} x(2n+1) W_N^{(2n+1)k} \qquad (3)$$

By utilizing the periodic nature of the rotation factor, it can be deduced that:

$$W_N^{2kn} = \left( W_{N/2}^{kn} \right), W_N^{(2n+1)k} = W_N^k W_{N/2}^{kn} \qquad (4)$$

Substituting Eq (4) into Eq (3) yields:

$$\begin{cases} X[k] = X_{\text{even}}[k] + W_N^k \cdot X_{\text{odd}}[k], \\ X[k+N/2] = X_{\text{even}}[k] - W_N^k \cdot X_{\text{odd}}[k], \end{cases} \qquad k = 0, 1, \ldots, N/2 - 1 \qquad (5)$$

Here, $X_{\text{even}}(k)$ represents the DFT computed for the even-indexed terms, and $X_{\text{odd}}(k)$ represents the DFT computed for the odd-indexed terms. In this manner, the computation of a DFT of length $N$ is split into two DFTs of length $N/2$, combined with a complex multiplication. This recursive process leads to direct computations when $N$ is reduced to 2.

## Squeeze-and-excitation attention mechanism

The attention mechanism mimics the selective processing of human beings to external information, aiming to focus on the key areas of input data, allowing the model to highlight important information adaptively.

The Squeeze-and-Excitation (SE) attention mechanism proposed by Hu et al. [32] recalibrates the importance of feature channels through global information squeezing (Squeeze) and channel-wise weight excitation (Excitation), thereby enhancing the network's ability to focus on key features. As shown in Fig 1, the SE mechanism consists of three steps: Squeeze, Excitation, and Scale.

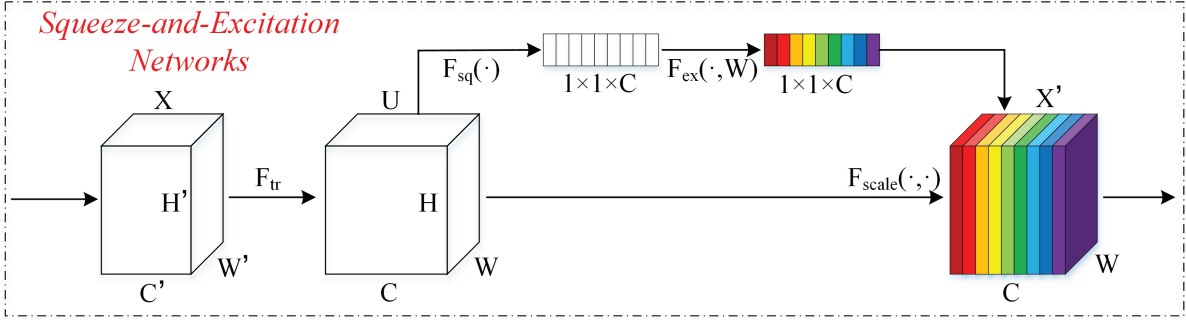

**Fig 1. The overall framework of the Squeeze-and-Excitation Attention Mechanism.**

Specifically, the Squeeze operation extracts the global spatial information from each channel through global pooling and produces a channel descriptor vector $Z_C$, which is computed as follows:

$$Z_C = F_{sq}(U_C) = \frac{1}{H \times W} \sum_{i=1}^{H} \sum_{j=1}^{W} U_C(i,j) \tag{6}$$

In Eq (6), $U_C$ signifies the c-th channel (with a dimension of $H \times W$) of the input feature map $U \in \mathbb{R}^{C \times H \times W}$. Subsequently, $Z_C$ is fed into the Excitation module, which consists of two fully connected layers. This module learns the intricate dependencies between channels through dimensionality reduction and expansion and produces the weight values $S$ for each channel. The computation is illustrated as follows:

$$S = F_{\text{ex}}(Z, W) = \sigma(W_2 \cdot \delta(W_1 \cdot Z)) \tag{7}$$

$W_1 \in \mathbb{R}^{\frac{C}{r} \times C}$ represents the first fully connected layer, which compresses the number of channels from $C$ to $\frac{C}{r}$. $W_2 \in \mathbb{R}^{C \times \frac{C}{r}}$ is the second fully connected layer responsible for restoring the number of channels back to $C$. $\delta$ and $\sigma$ denote the ReLU activation and Sigmoid functions, respectively. Finally, the Scale operation applies the learned channel weights to the original feature map, adjusting the importance of each channel. The formula is as follows:

$$X' = F_{\text{scale}}(S, U) = S \cdot U \tag{8}$$

$X'$ represents the feature map after $U$ has been adjusted with weights, enhancing the response of important feature channels while diminishing the impact of less significant channels.

## Cross-entropy loss function

In deep learning, the essence of training neural networks is optimizing their weight parameters to suit specific tasks, and the loss function serves as a powerful tool for updating the backpropagation and parameter adjustment of neural networks.

Most bearing fault diagnosis models based on deep learning use the cross-entropy loss function to measure the discrepancy between the model's predicted probability distribution and the true distribution. The cross-entropy loss [33] originates from information entropy and the Kullback-Leibler (KL) divergence, and its definition is as follows:

$$L_{CE} = -\sum_{i=1}^{C} y_i \log p_i \tag{9}$$

In Eq (9), $C$ denotes the total number of categories, $y_i$ represents the actual label of the $i$–$th$ sample (encoded with one-hot encoding), and $p_i$ indicates the probability that the model predicts the $i$–$th$ sample as belonging to the proper category.

During the model training process, by minimizing the cross-entropy loss, the discrepancy between the model's predicted probability distribution and the actual label distribution is gradually reduced, enhancing the model's classification accuracy and generalization capability.

## Proposed methods

This paper proposes a multi-domain collaborative denoising bearing fault diagnosis model based on a dynamic inter-domain attention mechanism (DIDAM) and noise-aware loss function (NALF), which utilizes both time-domain and frequency-domain signals as inputs, aiming to enhance the model's robustness and fault diagnosis performance in noisy environments.

### Multi-domain collaborative denoising diagnosis model

As shown in Fig 2, the proposed model consists of four core modules: data preprocessing, feature extraction, multi-domain feature fusion, and classification decision. Specifically, the components and their implementation steps are as follows:

**Data preprocessing:** Firstly, the bearing vibration signals (time-domain signals) collected by sensors are divided into several equal-length signal sequences of 1024. Then, these sequences are transformed into the frequency domain through the Fast Fourier Transform (FFT), and the data in both domains are normalized separately to eliminate dimensional discrepancies between different signals.

**Feature extraction:** A dual-branch parallel structure is constructed, with each branch containing five cascaded convolution-pooling operations. The time-domain branch focuses on extracting local temporal pattern features from vibration waveforms. In contrast, the frequency-domain branch aims to capture deep spectral energy distribution patterns, ultimately forming two sets of high-dimensional feature tensors.

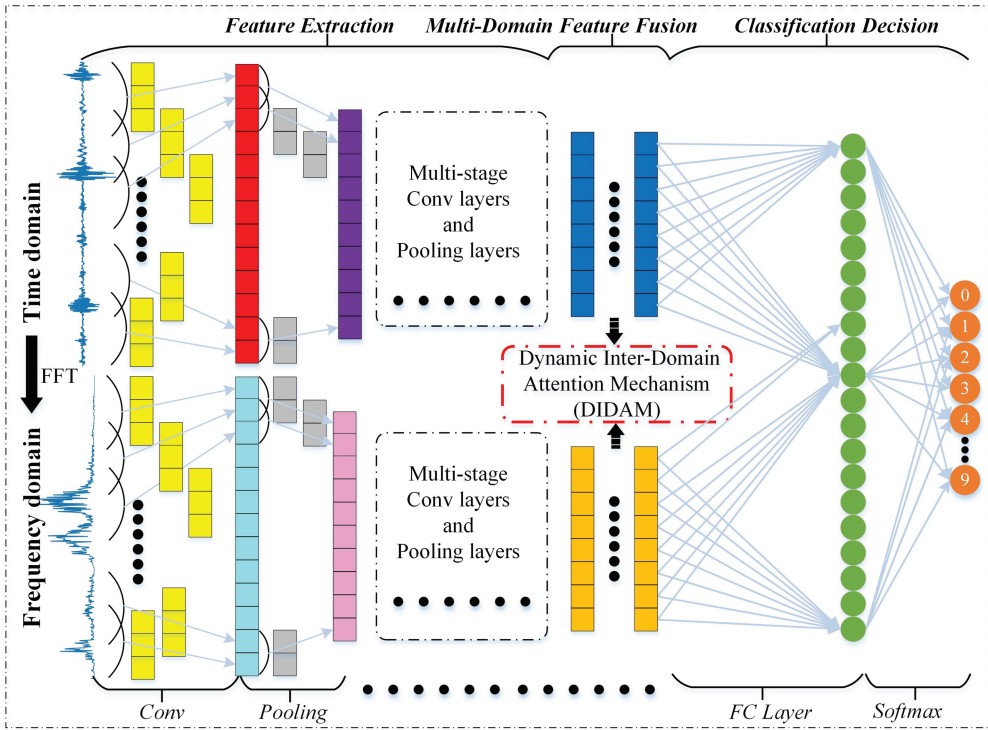

**Fig 2. The overall framework of the proposed multi-domain collaborative denoising bearing fault diagnosis model.**

**Multi-domain feature fusion:** To achieve efficient fusion of time-domain and frequency-domain information, a Dynamic Inter-Domain Attention Fusion mechanism (DIDAM) is designed. This mechanism can dynamically adjust the fusion method based on the feature importance of time-domain and frequency-domain signals, thus achieving flexible weighted fusion of information and enhancing the collaborative expression ability of cross-domain features.

**Classification decision:** The fused feature vectors are integrated through fully connected layers and then input into a softmax classifier for bearing fault signal classification. Moreover, to improve the model's applicability in industrial noise environments, a Noise-Aware Loss Function (NALF) is designed for model training and parameter updating, further enhancing classification accuracy.

For a more intuitive understanding of the specific architecture and parameter settings of the model proposed in this paper, Table 1 presents its detailed structure and parameter configuration.

## Dynamic inter-domain attention mechanism

A Dynamic Inter-Domain Attention Mechanism (DIDAM) is proposed to effectively capture key information in time-domain and frequency-domain signals and enhance the collaborative expression capability of cross-domain features. This mechanism is constructed by cascading an improved channel attention module with a novel spatial attention module, which dynamically weights the time-domain and frequency-domain features to achieve efficient information fusion.

In response to the Squeeze operation in SENet, which only employs global average pooling to extract channel descriptor vectors, leading to relatively monotonous information and limiting the diversity of feature channels and the comprehensiveness of information, we introduce global standard deviation pooling while retaining the original global average pooling. This aims to better aggregate channel information, thereby enhancing the expressive capability of the channel attention mechanism. The improved channel attention mechanism is shown in Fig 3, where $F_{sqd}(\cdot)$ denotes the newly added global standard deviation pooling operation. The mathematical expression for this operation is as follows:

**Table 1. Detailed architecture and parameter configuration of the proposed model.**

| | Time domain | | Frequency domain | |
|---|---|---|---|---|
| Layer type | Kernel size/stride | Kernel channel size | Kernel size/stride | Kernel channel size |
| Convolution 1 | 64×1/1×1 | 16 | 64×1/1×1 | 16 |
| Pooling 1 | 16×1/16×1 | 16 | 16×1/16×1 | 16 |
| Convolution 2 | 32×1/1×1 | 64 | 32×1/1×1 | 64 |
| Pooling 2 | 2×1/2×1 | 64 | 2×1/2×1 | 64 |
| Convolution 3 | 32×1/1×1 | 128 | 32×1/1×1 | 128 |
| Pooling 3 | 2×1/2×1 | 128 | 2×1/2×1 | 128 |
| Convolution 4 | 16×1/1×1 | 128 | 16×1/1×1 | 128 |
| Pooling 4 | 2×1/2×1 | 128 | 2×1/2×1 | 128 |
| Convolution 5 | 16×1/1×1 | 128 | 16×1/1×1 | 128 |
| Pooling 5 | 2×1/2×1 | 128 | 2×1/2×1 | 128 |
| Dynamic inter-domain attention mechanism | | | | |
| Fully-connected | 100 | 1 | - | - |
| Softmax | 10 | 2 | - | - |

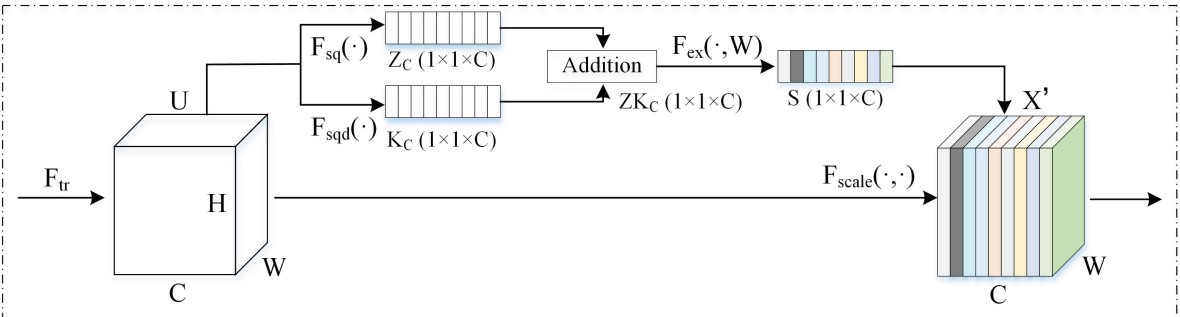

**Fig 3. The schematic diagram of the channel attention module in Dynamic Inter-Domain Attention Mechanism (DIDAM).**

$$K_C = F_{sqd}(U_C) = \sqrt{\frac{1}{H \times W} \sum_{i=1}^{H} \sum_{j=1}^{W} (U_C(i,j) - Z_C)^2} \tag{10}$$

Next, the sum operation is performed on $K_C$ and $Z_C$ to obtain the final channel description vector $ZK_C$. The other operations remain consistent with SENet and will not be altered. The formula is as follows:

$$ZK_C = K_C \oplus U_C \tag{11}$$

In Eq (11), the symbol $\oplus$ denotes the vector summation operation. Although traditional spatial attention mechanisms can prompt the model to focus on key areas within the feature map, their fixed convolutional windows or weighting mechanisms may overlook the nonlinear relationships between regions, failing to attend to the subtle associations between different areas effectively.

Therefore, this paper proposes a novel spatial attention mechanism, as illustrated in Fig 4. Firstly, three 1×1 convolution operations are performed on the channel-weighted feature map

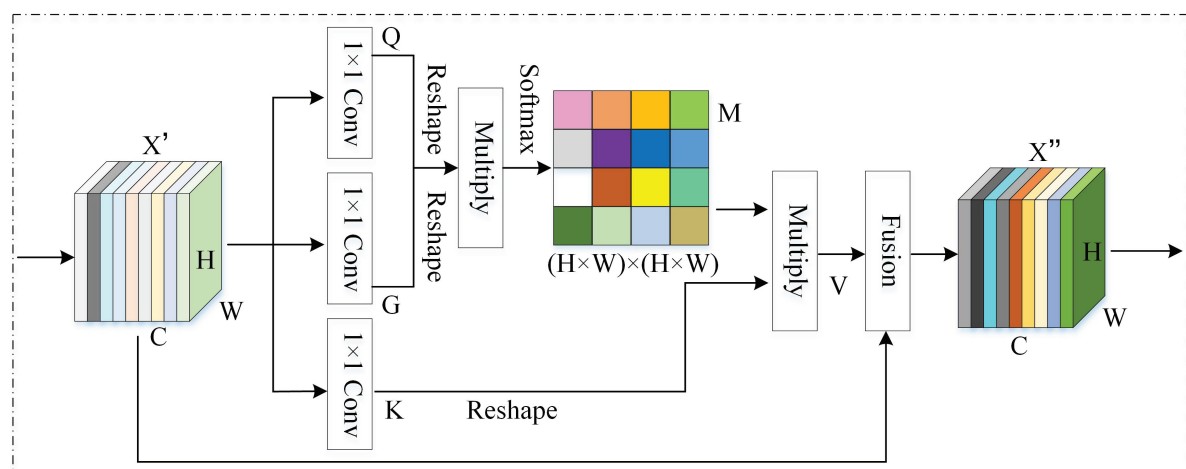

**Fig 4. The schematic diagram of the spatial attention module in Dynamic Inter-Domain Attention Mechanism (DIDAM).**

$X' \in \mathbb{R}^{C \times H \times W}$, resulting in three feature maps: $Q \in \mathbb{R}^{C \times H \times W}$, $G \in \mathbb{R}^{C \times H \times W}$, and $K \in \mathbb{R}^{C \times H \times W}$. The formulas are as follows:

$$Q = f^{1 \times 1}(X'), G = f^{1 \times 1}(X'), K = f^{1 \times 1}(X') \tag{12}$$

Next, the $Q$, $G$, and $K$ feature maps are each subjected to a reshape operation to obtain $Q \in \mathbb{R}^{HW \times C}$, $G \in \mathbb{R}^{C \times HW}$, and $K \in \mathbb{R}^{C \times HW}$. The matrix multiplication is then performed between $Q$ and $G$, and the resulting weights are normalized using the Softmax function to obtain $M \in \mathbb{R}^{(H \times W) \times (H \times W)}$, which represents the spatial attention map. The formula is as follows:

$$M_{ji} = \frac{\exp(Q_i, G_j)}{\sum_{i=1}^{N} \exp(Q_i, G_j)} \tag{13}$$

In Eq (13), a higher value of $M$ indicates a more remarkable similarity between the two regions. The obtained $M$ is multiplied by $K$, and the resulting product is reshaped into a new feature map of dimensions $V \in \mathbb{R}^{C \times H \times W}$.

Finally, $V$ is fused with the input feature $X'$ to obtain the final output feature map $X''$. The mathematical formula is as follows:

$$X_j'' = \sum_{i=1}^{N} \left( M_{ji} K_i \right) + X_j' \tag{14}$$

In summary, by reshaping the feature maps and performing multiple matrix operations on the similarities between regions, the spatial attention module in DIDAM can more effectively handle the nonlinear relationships between regions. Thus, it can capture more complex spatial dependencies and avoid the issue of neglecting fine-grained spatial information in traditional methods.

## Noise-aware loss function

When used for bearing fault diagnosis, the cross-entropy loss assigns the same loss weight to all samples without considering their difficulty level. This approach may lead to the cumulative loss of many simple samples, overshadowing the contribution of complex samples and causing the gradient updates to be biased towards simple samples. Additionally, the cross-entropy loss is sensitive to outliers. When there are outliers or noise in the dataset, the model may overly focus on these points, thus affecting its generalization ability.

Therefore, this paper proposes a Noise-Aware Loss Function (NALF). Specifically, since the performance of deep learning models becomes gradually robust with the increase in training epochs, a threshold $\tau$ is introduced, which increases as the number of training epochs increases. The formula is as follows:

$$\tau = \frac{1}{N} + \left(1 - \frac{1}{N}\right) / E_s * E \tag{15}$$

In Eq (15), $N$ represents the number of sample categories, $E_s$ represents the total number of training epochs, and $E$ represents the current training epoch. During each training epoch, the model compares the probability value of each sample being predicted correctly with the threshold $\tau$. If the value is greater than $\tau$, it is considered a correct prediction and is assigned a lower weight; otherwise, it is assigned a higher weight. The formula is as follows:

$$\begin{cases} W_l = (1 - P_r)^2, & \text{if } P_r > \tau \\ W_h = 1, & \text{otherwise} \end{cases} \tag{16}$$

Here, $P_r$ represents the probability value the model correctly predicts each sample in the current round. $W_l$ and $W_h$ denote the lower and higher weights, respectively, with their values set according to reference [34]. The mathematical expression for the Noise-Aware Loss Function (NALF) is as follows:

$$L_{\text{NALF}} = \begin{cases} L_{CE} \times W_l, & \text{if } P_r > \tau \\ L_{CE} \times W_h, & \text{otherwise} \end{cases} \tag{17}$$

This design enables the model to update its parameters based on the actual prediction for each sample. In other words, it can effectively reduce the influence of prediction errors caused by noise. When the model's predictions are correct, it learns more reliable decision rules, thereby mitigating the adverse effects of noise on the model.

## Experimental validation and results discussion

### Experimental setup and noise simulation

All experiments in this chapter were conducted on a computing platform based on the Windows 11 operating system, equipped with a 12th Generation Intel®Core™ processor and an NVIDIA RTX 4090 graphics card. Algorithm development was implemented in Python, relying on the PyCharm integrated development environment and the PyTorch deep learning framework. During the model training process, the Adam optimizer was used, with an initial learning rate set to 0.001, a batch size of 128, and a total of 100 epochs to ensure the stability and efficiency of model convergence.

Given that Gaussian white noise is statistically similar to various random noises in natural environments, the experiments will be conducted against backgrounds of Gaussian white noise with different intensities. Specifically, this study selects six different Signal-to-Noise Ratio (SNR) levels, including –10 dB, –8 dB, –6 dB, –4 dB, –2 dB, and 0 dB, to evaluate the performance of the proposed model under varying noise conditions. The formula for calculating the SNR is as follows:

$$\text{SNR}_{dB} = 10 \log_{10} \frac{P_s}{P_n} \tag{18}$$

Here, $P_s$ and $P_n$ represent the signal and noise power, respectively, and $dB$ denotes decibels. A higher Signal-to-Noise Ratio (SNR) indicates that the proportion of the signal relative to the noise is greater, meaning the signal is purer; conversely, a lower SNR suggests a higher proportion of noise, and the signal is more severely interfered with.

### Dataset srescription

**CWRU Dataset** [35]: This dataset was collected at a sampling rate of 12 kHz under four load conditions (0, 1, 2, and 3 horsepower). It includes four types of faults: healthy (H), inner race fault (IF), outer race fault (OF), and rolling element fault (BF). Each fault type is further categorized into three damage sizes: 0.007, 0.014, and 0.021 inches. Details of the CWRU dataset used in this study are shown in Table 2.

**Table 2. Details of the CWRU dataset used in this study.**

| Fault location | Labels | Load | Diameter/ Sample frequeney | Sampling points | Training set size | Test set size |
|---|---|---|---|---|---|---|
| Roller fault | 0 | 0 hp | 0.007 inch | 1024 | 300 | 100 |
| | 1 | 0 hp | 0.014 inch | 1024 | 300 | 100 |
| | 2 | 0 hp | 0.021 inch | 1024 | 300 | 100 |
| Inner race | 3 | 0 hp | 0.007 inch | 1024 | 300 | 100 |
| | 4 | 0 hp | 0.014 inch | 1024 | 300 | 100 |
| | 5 | 0 hp | 0.021 inch | 1024 | 300 | 100 |
| Outer race | 6 | 0 hp | 0.007 inch | 1024 | 300 | 100 |
| | 7 | 0 hp | 0.014 inch | 1024 | 300 | 100 |
| | 8 | 0 hp | 0.021 inch | 1024 | 300 | 100 |
| Normal | 9 | 0 hp | - | 1024 | 300 | 100 |

**MFPT Dataset** [36]: This dataset consists of 23 categories. The basic group includes three sets of normal and three sets of outer race fault data collected under a load of 270 lbs, a shaft speed of 25 Hz, and a sampling rate of 97656 kHz. The extended group contains two variable-load subsets: seven sets of outer race fault data (collected at loads ranging from 25 to 300 lbs with a sampling rate of 48828 kHz) and seven sets of inner race fault data (collected at loads ranging from 0 to 300 lbs with a sampling rate of 48828 kHz). Details of the MFPT dataset used in this study are shown in Table 3.

## Fault diagnosis under different noise environments

This subsection analyzes the fault diagnosis performance of the proposed model under different signal-to-noise ratio (SNR) conditions. As shown in Fig 5(a), on the CWRU dataset, the diagnosis accuracy reaches 100% when SNR = 0 dB. The accuracy gradually declines as the SNR decreases (i.e., noise increases). However, it remains above 96.91% for SNR $\geq$ –4 dB, indicating that the model exhibits strong noise resistance in high-SNR environments. When the SNR drops to –6 dB and –8 dB, the accuracy decreases to 94.58% and 89.62%, respectively, demonstrating that despite increased noise interference, the model can still accurately identify various fault types. Even in an extreme noise environment (SNR = –10 dB), the accuracy remains at 81.25%, proving that the model can effectively extract fault features under severe noise conditions, further highlighting its superior noise robustness.

As shown in Fig 5(b), on the MFPT dataset, the proposed model achieves a fault diagnosis accuracy of 96.74% at SNR = 0 dB, slightly lower than the 100% accuracy observed on the

**Table 3. Details of the MFPT dataset used in this study.**

| Fault location | Labels | Load | Diameter/Sample frequeney | Sampling points | Training set size | Test set size |
|---|---|---|---|---|---|---|
| Inner race | 0 | 0 lbs | 48828 Hz | 1024 | 300 | 100 |
| | 1 | 50 lbs | 48828 Hz | 1024 | 300 | 100 |
| | 2 | 150 lbs | 48828 Hz | 1024 | 300 | 100 |
| | 3 | 300 lbs | 48828 Hz | 1024 | 300 | 100 |
| Outer race | 4 | 0 lbs | 48828 Hz | 1024 | 300 | 100 |
| | 5 | 50 lbs | 48828 Hz | 1024 | 300 | 100 |
| | 6 | 150 lbs | 48828 Hz | 1024 | 300 | 100 |
| | 7 | 270 lbs | 48828 Hz | 1024 | 300 | 100 |
| | 8 | 300 lbs | 48828 Hz | 1024 | 300 | 100 |
| Normal | 9 | 270 lbs | 97656 Hz | 1024 | 300 | 100 |

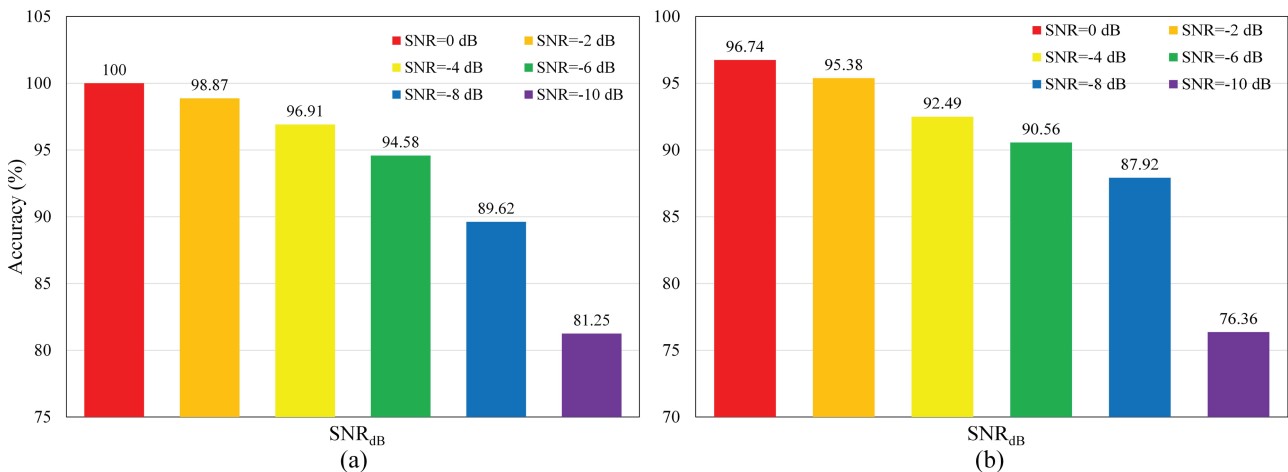

**Fig 5. The fault diagnosis performance of the proposed model under different SNR conditions on the test datasets of CWRU and MFPT. (a) CWRU dataset; (b) MFPT dataset.**

CWRU dataset. As the SNR decreases, the fault recognition rate also declines. However, it remains above 90% for SNR ≥ −6 dB, demonstrating that the model can still stably identify fault features under moderate noise levels. Even in the extreme case of SNR = −10 dB, the accuracy remains at 76.36%, significantly higher than random guessing, indicating that the model can still effectively extract fault features and maintain strong noise resistance. Additionally, compared to the CWRU dataset, the accuracy of the MFPT dataset is slightly lower under the same SNR conditions, which may be attributed to the lower signal quality and more significant background noise interference in the MFPT dataset. Notably, in both datasets, the accuracy of the proposed model does not exhibit a drastic drop, suggesting that it can maintain stable performance not only in high-SNR environments but also under low-SNR conditions, fully demonstrating its superior noise robustness.

Additionally, we plotted the confusion matrix to visually present the model's recognition of different fault types under various SNR conditions. Since the accuracy of the model on the CWRU dataset reached 100% when SNR = 0 dB, the confusion matrix is not shown for this case. As shown in Fig 6, when SNR ≥ −6 dB, most misclassifications occur in Category 2, which is often misidentified as Category 0 or Category 4. This suggests that the features of Category 2 lack sufficient discriminative power under low SNR conditions, causing its features to spread across multiple categories. When the SNR further drops to −8 dB, in addition to the misclassification of Category 2, Category 0 is also misidentified as Category 2, indicating that the features of Category 0 and Category 2 become more similar under the influence of noise. This may be due to their close time-frequency features, with the noise masking their subtle differences. When the SNR decreases to −10 dB, almost all categories experience severe misclassification, indicating that extreme noise leads to extreme instability in category features, further weakening the model's discriminative ability.

To further validate this phenomenon, we performed t-SNE dimensionality reduction and visualization analysis on the feature distributions under different SNR conditions, as shown in Fig 7. The results indicate that, on the CWRU dataset, the clustering of samples from different categories is relatively good under high SNR conditions. As the SNR decreases, the distribution of points becomes increasingly dispersed, and the category boundaries become

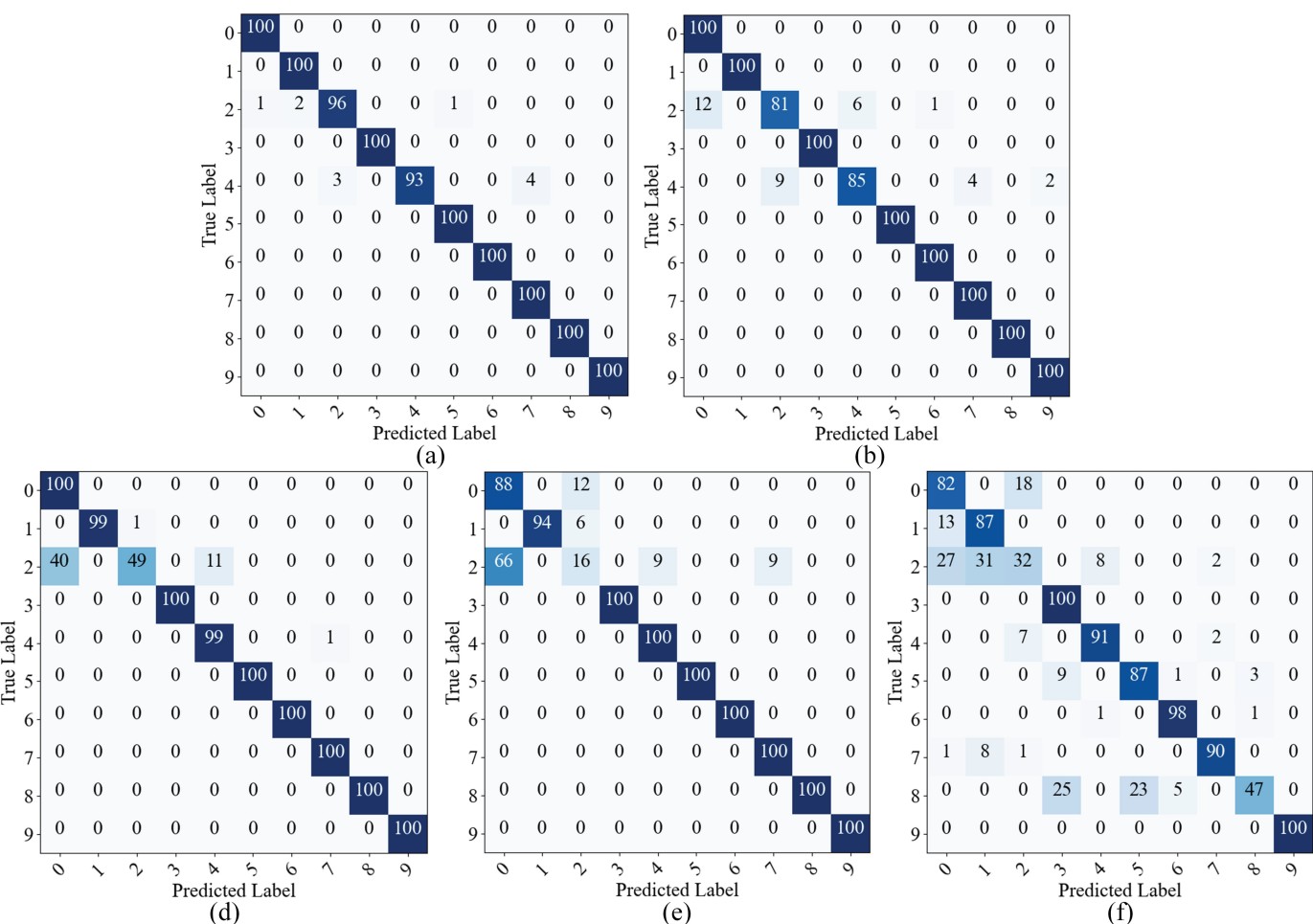

**Fig 6. The confusion matrix of the proposed model under different SNR conditions on the test dataset of CWRU. (a) −2 dB; (b) −4 dB; (c) −6 dB; (d) −8 dB; (e) −10 dB.**

more blurred. Specifically, when SNR ≥ −8 dB, Categories 0, 2, and 4 gradually move closer in the t-SNE plot and eventually overlap. When SNR = −10 dB, most category sample points intersect, making the categories nearly inseparable. This trend corresponds with the misclassification results from the confusion matrix, further confirming the impact of noise on fault feature extraction.

On the MFPT dataset, we also plotted the confusion matrix and t-SNE dimensionality reduction visualization results (Figs 8 and 9). The confusion matrix shows that when SNR ≥ −4 dB, the main misclassification occurs between Categories 0, 1, and 2, indicating that these categories have similar features under low SNR conditions. When the SNR drops to −6 dB and below, misclassifications between Categories 0, 1, 2, 4, and 5 significantly increase, suggesting that as noise interference intensifies, the feature distinguishability between different categories decreases. From the t-SNE dimensionality reduction visualization, compared to the CWRU dataset, the distribution of sample points for each category on the MFPT dataset is more dispersed, further confirming that the signal quality of the MFPT dataset is poorer and that the noise impact is more significant.

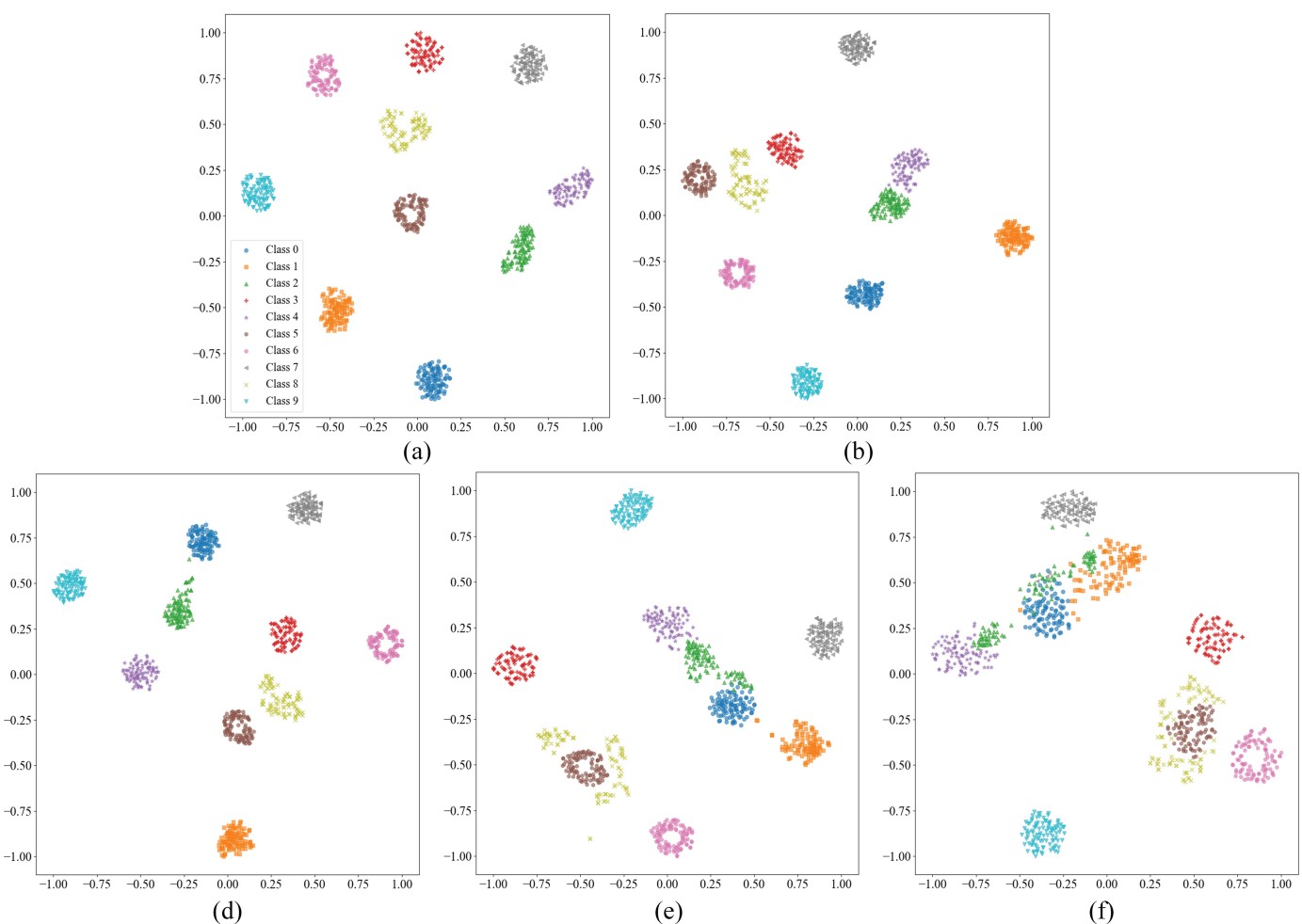

**Fig 7. The t-SNE visualization of feature distribution under different SNR conditions on the test dataset of CWRU. (a) −2 dB; (b) −4 dB; (c) −6 dB; (d) −8 dB; (e) −10 dB.**

## Comparison with the state-of-the-art methods

To further verify the noise resistance of the proposed model, we compared its performance with six state-of-the-art denoising models (including MCNN-LSTM [24], AMFCN [26], Mel-CNN [37], DRSN [38], IDRSN [39], and RDDAN [40]) under different SNR conditions, as shown in Table 4. As the SNR decreases (i.e., noise level increases), the diagnostic accuracy of all models declines—an expected trend commonly observed in denoising tasks. Specifically, under high SNR conditions (SNR ≥ −4 dB), the proposed model outperforms most existing methods, ranking only behind RDDAN and AMFCN. In the lower SNR range (−4 ≤ SNR ≤ −10 dB), its diagnostic accuracy is second only to RDDAN. In boisterous environments (SNR = −8 dB and SNR = −10 dB), the performance advantage of the proposed model becomes more pronounced. It is emphasized that the proposed model shows the least degradation in performance as the SNR ratio continues to decrease, fully demonstrating its superior noise robustness. In summary, the proposed model delivers outstanding fault diagnosis performance under various noise levels, particularly with extreme robustness in high-noise environments, indicating its high practical value for real-world applications.

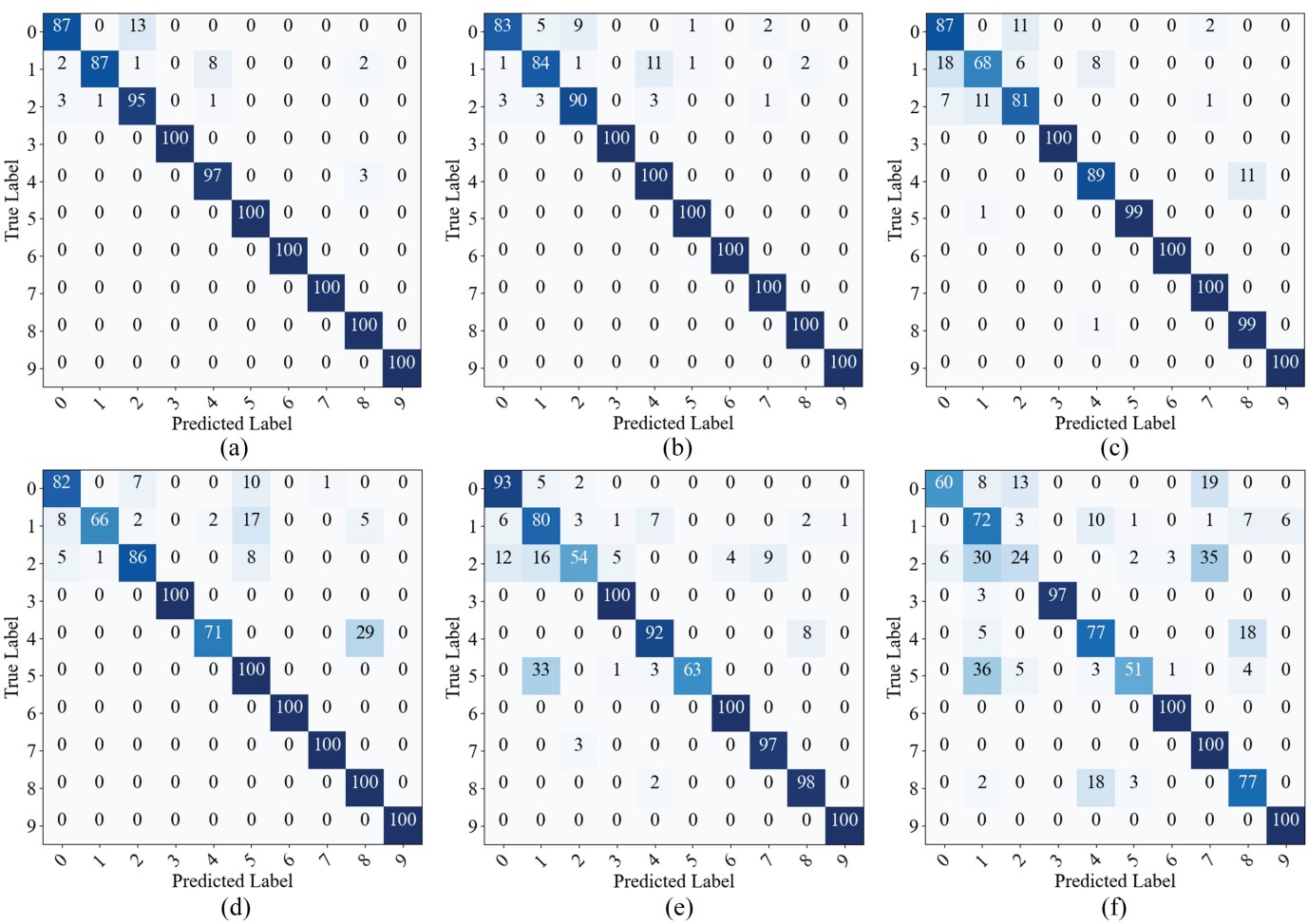

**Fig 8. The confusion matrix of the proposed model under different SNR conditions on the test dataset of MFPT. (a) 0 dB; (b) −2 dB; (c) −4 dB; (d) −6 dB; (e) −8 dB; (f) −10 dB.**

To more intuitively demonstrate the performance differences among various models under strong noise conditions, we selected SNR = −8 dB ( high noise intensity and covers a comprehensive range of comparison methods) to plot their confusion matrices and t-SNE dimensionality reduction visualizations on the CWRU test set, as shown in Figs 10 and 11.

From the confusion matrices, it is evident that models such as DRSN, AMFCN, Mel-CNN, and IDRSN have significant classification errors across multiple categories, primarily between categories 1, 2, 5, and 8. This may be due to the limited noise resistance of these methods, which makes it difficult to effectively distinguish between these inherently similar fault patterns under high noise interference. In contrast, our model only has a few misclassifications on categories 0, 1, and 2, with most errors being the misidentification of category 2 as category 0. This phenomenon may stem from the substantial similarity between categories 0 and 2 in the feature space, which is more easily obscured by intense noise. It is worth noting that although the overall recognition accuracy of the proposed model is slightly lower than that of RDDAN, its misclassifications are mainly between adjacent categories and are relatively concentrated, which to some extent reflects its strong ability to model feature boundaries and maintain high fault recognition capabilities even in high-noise environments.

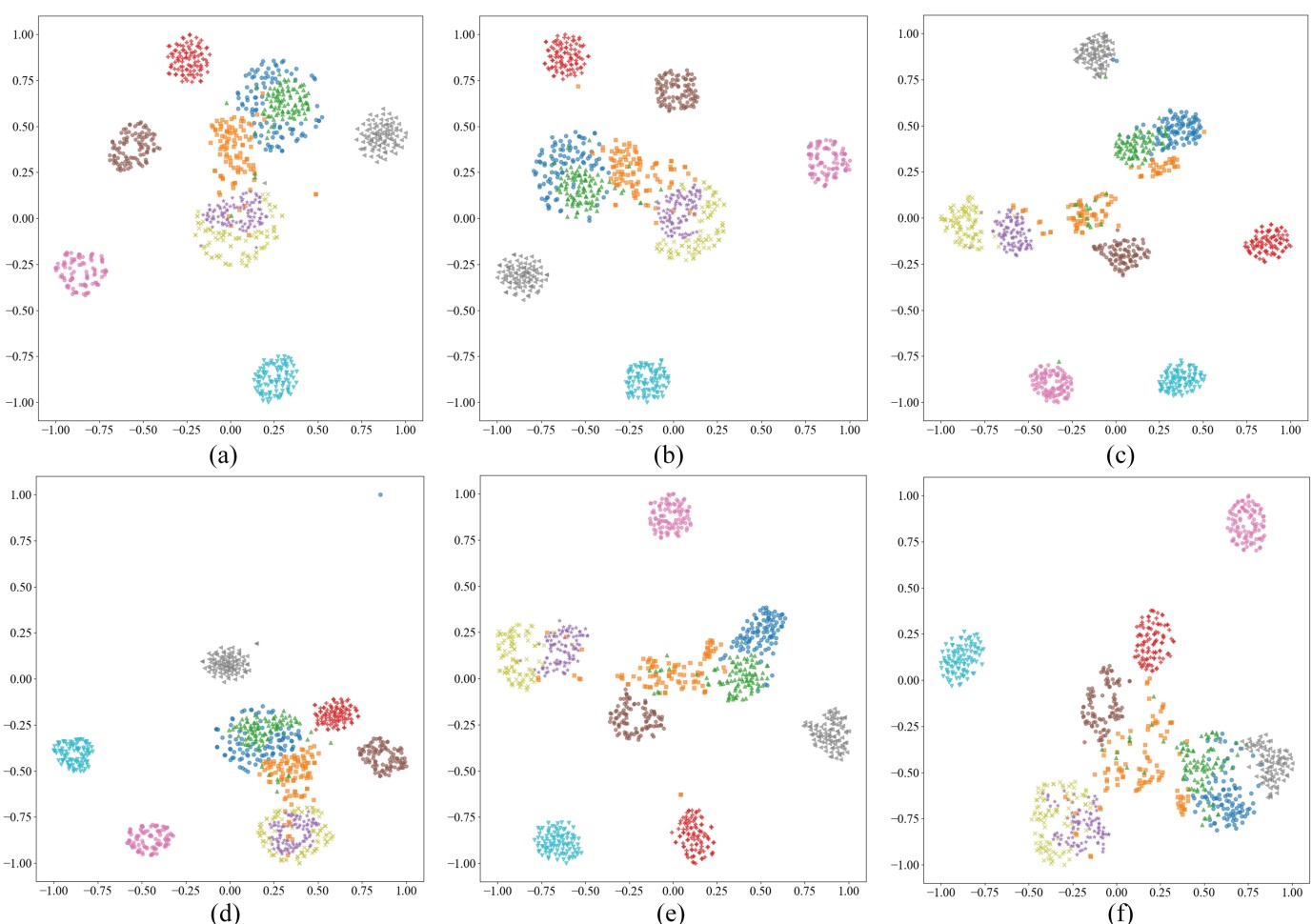

**Fig 9. The t-SNE visualization of feature distribution under different SNR conditions on the test dataset of MFPT. (a) 0 dB; (b) −2 dB; (c) −4 dB; (d) −6 dB; (e) −8 dB; (f) −10 dB.**

**Table 4. Comparison of fault diagnosis performance of various methods under different signal-to-Noise ratio conditions on CWRU dataset.**

| Model | 0 dB | −2 dB | −4 dB | −6 dB | −8 dB | −10 dB |
|---|---|---|---|---|---|---|
| DRSN [38] (2019) | 98.9% | 97.2% | 89.2% | 82.4% | 79.0% | 68.6% |
| MCNN-LSTM [24] (2021) | 81.41% | 77.27% | – | – | – | – |
| RDDAN [40] (2022) | 100% | 99.7% | 98.4% | 97.8% | 92.5% | 85.2% |
| AMFCN [26] (2023) | 99.82% | 99.51% | 99.16% | 93.55% | 81.92% | – |
| Mel-CNN [37] (2023) | 99.5% | 97.2% | 94.1% | 88.6% | 81.2% | 72.4% |
| IDRSN [39] (2023) | 99.6% | 98.5% | 92.7% | 86.3% | 82.7% | 76.4% |
| Ours | 100.00% | 98.87% | 96.91% | 94.58% | 89.62% | 81.25% |

The t-SNE dimensionality reduction visualization results shown in Fig 11 further corroborate this conclusion. It is visible that DRSN, AMFCN, Mel-CNN, and IDRSN have significant overlapping regions among categories 1, 2, 5, and 8, indicating their difficulty in achieving adequate category distinction under strong noise. In contrast, the feature

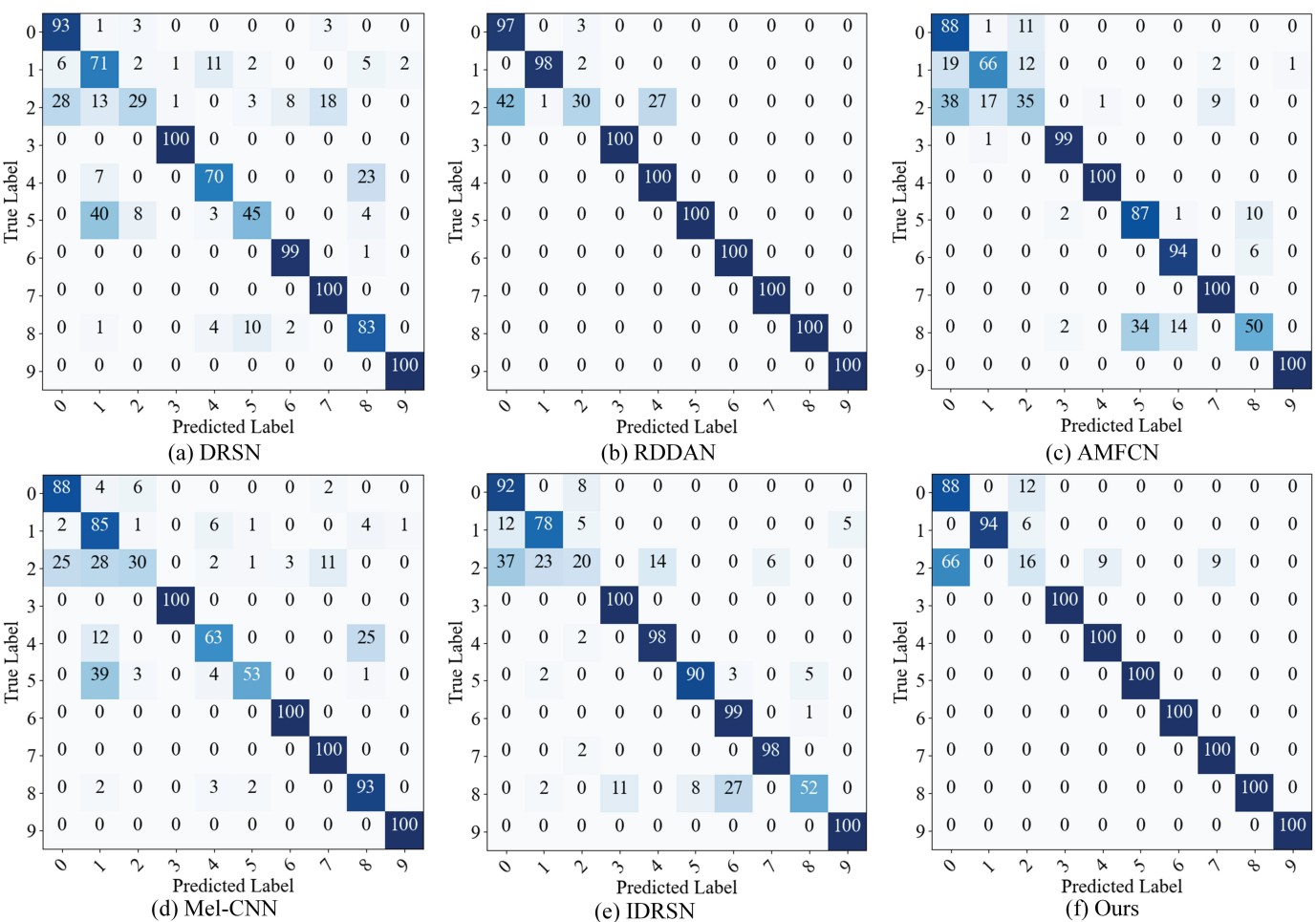

**Fig 10. The comparison of confusion matrices of different models on the CWRU test set under SNR = −8 dB.**

distributions extracted by our method and RDDAN are much clearer, with distinct boundaries between different categories and almost no overlap, which is highly consistent with the confusion matrix results. This further demonstrates the proposed model's superior feature representation capability and good inter-class separability under complex noise conditions.

## Ablation experiment

Finally, to analyze the impact of the Dynamic Inter-domain Attention Mechanism (DIDAM) and the Noise-Aware Loss Function (NALF) on the performance of the proposed model, we conducted an ablation study on the CWRU dataset, as shown in Fig 12. All models in this experiment follow the framework described in the "Multi-Domain Collaborative Denoising Diagnosis Model," with the specific experimental settings as follows:

**Model 1**: DIDAM is not used, and the loss function is cross-entropy.
**Model 2**: DIDAM is used, and the loss function is cross-entropy.
**Model 3**: DIDAM is not used, and the loss function is NALF.
**Model 4**: The proposed model (DIDAM + NALF).

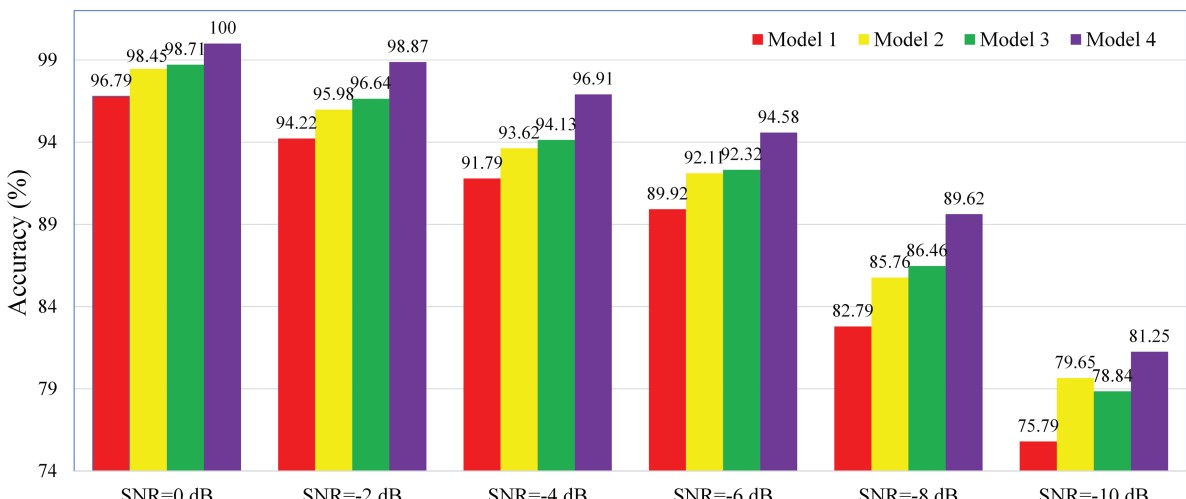

**Fig 11. The t-SNE visualization of feature distributions of different models on the CWRU test set under SNR = −8 dB.**

**Fig 12. Impact of DIDAM and NALF on the fault diagnosis performance of the proposed model on the CWRU dataset.**

The experimental results show that Model 2 outperforms Model 1 under all SNR conditions, particularly at low SNR levels (SNR = –8 dB and SNR = –10 dB), demonstrating that DIDAM significantly enhances the model's robustness. Similarly, Model 3 performs better than Model 1 in most SNR conditions, especially in low-SNR environments (e.g., SNR = –8 dB and SNR = –10 dB), indicating that NALF helps improve the model's noise resistance. Notably, Model 4 surpasses all three baseline models across all SNR conditions, achieving an accuracy of 81.25% even in extreme noise conditions (SNR = –10 dB). This result confirms that combining DIDAM and NALF can significantly enhance the model's noise robustness and overall diagnostic accuracy. In summary, both DIDAM and NALF contribute to performance improvement through different mechanisms, and their combination further amplifies this effect, resulting in a synergistic enhancement.

## Discussion

Noise interference in industrial scenarios often masks the original characteristics of bearing fault signals, leading to insufficient diagnostic reliability of fault diagnosis models based on deep learning. Although existing denoising models have achieved some success, they still have certain limitations:

- Most methods focus exclusively on processing time-domain signals, neglecting key information contained in other potential domains. This makes the model's accuracy in extracting the original features of fault signals under intense background noise difficult, limiting its ability to express high-dimensional features of different fault patterns.
- Some methods have begun to attempt feature extraction from different domains. However, they typically rely on fixed feature concatenation or fusion methods, failing to fully consider the differential contributions of features from different signal domains to model performance, leading to poor multi-source information fusion.
- Existing denoising models usually employ standard cross-entropy functions for network parameter updates, neglecting that strong noise environments may lead the model to make incorrect decisions, causing deviations in the direction of gradient updates.

This paper proposes a multi-domain collaborative denoising bearing fault diagnosis model based on a dynamic inter-domain attention mechanism and a noise-aware loss function. The model can simultaneously extract high-dimensional features of bearing fault signals from different domains, enriching and diversifying the expression of the original signal's high-dimensional features and enhancing the model's fault recognition capability under substantial noise interference. Additionally, a dynamic inter-domain attention mechanism is designed to flexibly adjust the fusion approach according to the importance of each signal domain, improving the efficiency and accuracy of information fusion. Finally, a noise-aware loss function is constructed, effectively avoiding deviations in the model's gradient update direction caused by strong noise environments, thereby improving the stability and reliability of fault diagnosis.

Experimental results on the publicly available CWRU and MFPT datasets show that the proposed model exhibits superior diagnostic capabilities under different noise intensities. Even in an extreme noise environment with an SNR of –10 dB, the model achieved 81.25% and 76.36% diagnostic accuracy rates. Compared to most existing mainstream denoising models, the proposed model maintains high diagnostic precision under low SNR conditions (extreme noise environments), fully demonstrating its effectiveness.

Despite the promising experimental results of the proposed model, there are still some limitations. For example, the experiments in this paper are based on standardized datasets

such as CWRU and MFPT. In contrast, the complex conditions in actual industrial settings (such as variable speed, variable load, and multi-source mixed noise) may lead to stronger time-variability in fault signals, resulting in decreased model generalization ability. The multi-domain feature extraction and dynamic inter-domain attention mechanism increase the computational burden, and the model may struggle to meet real-time diagnostic requirements in resource-constrained scenarios such as industrial edge devices. Moreover, the model relies on training data with known fault types and may perform poorly on unseen fault patterns (such as complex or early weak faults), lacking incremental learning capability.

In the future, a lightweight and more practical denoising bearing fault diagnosis model can be designed to reduce computational latency while maintaining precision. Additionally, introducing non-Gaussian noise, multi-source mixed noise, and varying condition data can enhance the model's environmental adaptability. Furthermore, meta-learning or contrastive learning can be combined to improve the model's ability to identify new faults with a small number of samples.

## Conclusion

This paper proposes a multi-domain collaborative denoising bearing fault diagnosis model based on the dynamic inter-domain attention mechanism and noise-aware loss function. Specifically, the model can simultaneously extract high-dimensional features of bearing fault signals from different domains, aiming to enrich and diversify the expression of high-dimensional features in the original signals, thereby enhancing the model's fault recognition ability under substantial noise interference. Secondly, a dynamic inter-domain attention mechanism (DIDAM) is introduced to effectively distinguish the importance of information from different signal domains and enhance the collaborative expression ability of cross-domain features. Lastly, a noise-aware loss function (NALF) is proposed for model training and parameter updating. It aims to effectively reduce the negative impact of noise on the model, allowing it to learn more reliable decision-making rules. Experimental results on two public datasets, CWRU and MFPT, show that compared to most existing mainstream denoising fault diagnosis models, the proposed model exhibits superior diagnostic capabilities under different noise intensities. Particularly in extreme noise environments, its anti-interference ability and robustness are effectively improved, demonstrating its effectiveness.

## Author contributions

**Data curation:** Weilin Cao.

**Funding acquisition:** Weilin Cao.

**Methodology:** Weilin Cao.

**Resources:** Weilin Cao.

**Software:** Weilin Cao, Liqiang Zhang.

**Supervision:** Weilin Cao, Liqiang Zhang.

**Validation:** Liqiang Zhang.

**Writing – original draft:** Weilin Cao, Liqiang Zhang.

**Writing – review & editing:** Weilin Cao, Liqiang Zhang.

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
