## [Decision Letter · Decision Letter 0]

1 Apr 2025

PONE-D-25-14574A Multi-Domain Collaborative Denoising Bearing Fault Diagnosis Model Based on Dynamic Inter-Domain Attention Mechanism and Noise-Aware Loss FunctionPLOS ONE

Dear Dr. Cao,

Thank you for submitting your manuscript to PLOS ONE. After careful consideration, we feel that it has merit but does not fully meet PLOS ONE’s publication criteria as it currently stands. Therefore, we invite you to submit a revised version of the manuscript that addresses the points raised during the review process.

We look forward to receiving your revised manuscript.

Kind regards,

Kannadhasan Suriyan

Academic Editor

PLOS ONE

“This research was supported by Research and Innovation Team Project of Neijiang Normal University (No. 18TD02).”

“This research was supported by Research and Innovation Team Project of Neijiang Normal University(No. 18TD02).”

“This research was supported by Research and Innovation Team Project of Neijiang Normal University (No. 18TD02).”

Additional Editor Comments:

Author should submit the revised paper as per reviewer comments

Reviewers' comments:

Reviewer's Responses to Questions

**Comments to the Author**

1. Is the manuscript technically sound, and do the data support the conclusions?

Reviewer #1: Yes

Reviewer #2: Yes

2. Has the statistical analysis been performed appropriately and rigorously? 

Reviewer #1: Yes

Reviewer #2: Yes

3. Have the authors made all data underlying the findings in their manuscript fully available?

Reviewer #1: Yes

Reviewer #2: Yes

4. Is the manuscript presented in an intelligible fashion and written in standard English?

Reviewer #1: Yes

Reviewer #2: Yes

5. Review Comments to the Author

Reviewer #1: This work proposes a multi-domain collaborative denoising bearing fault diagnosis model based on dynamic inter-domain attention mechanism and noise-aware loss. The paper is well-structured and presents promising results. To further enhance the manuscript, the following suggestions are offered:

1.The Abstract could benefit from a clearer emphasis on the practical significance of this research. Highlighting the engineering context and potential real-world applications would help readers better understand the value of the proposed method.

2.While the authors have provided a thorough review of current research, it would be beneficial to more explicitly summarize the existing research gaps before introducing the contributions and novelty of this work. Additionally, expanding the discussion to include emerging trends in machine learning and signal processing—particularly in industrial applications—could strengthen the manuscript. For example, exploring connections to recent advancements in areas such as digital twin methodology for vibration-based monitoring and prediction of gear wear, novel cyclic-correntropy based indicator for gear wear monitoring, and neuro-fuzzy system-guided cross-modal zero-sample diagnostic framework using multi-source heterogeneous non-contact sensing data could provide valuable context and demonstrate the broader relevance of this research.

3.The resolution of the figures in the manuscript could be improved to ensure clarity and enhance the overall presentation of the results.

4.There are occasional grammatical errors throughout the manuscript. A thorough proofreading to address these issues would improve the readability and professionalism of the paper.

5.Including a section on potential future research directions at the end of the Conclusion would provide a forward-looking perspective and inspire further exploration in this field.

Reviewer #2: 1.The manuscript is generally clear and conveys the research content and methodology effectively. However, there are several grammatical errors, overly complex sentence structures, and awkward phrasing that affect readability. Some technical expressions are rigid, making it difficult for readers to follow the core ideas. I recommend further language polishing to enhance fluency and readability, ensuring that the content is easily understandable.

2.Your study presents certain improvements and new approaches, particularly in signal processing and fault diagnosis within the field of mechatronic engineering. However, the level of innovation is not sufficiently emphasized. The differences between your proposed method and existing approaches need to be highlighted more clearly. I suggest strengthening the comparison in the introduction and discussion sections to explicitly outline the novel contributions of your work.

3.Your manuscript generally aligns with the journal’s scope, which covers topics related to mechatronic engineering. However, the experimental validation lacks strong supporting evidence. The manuscript would benefit from additional comparative experiments or a more detailed performance analysis. To improve the scientific rigor of the study, I strongly recommend providing more experimental data, especially direct comparisons with existing methods.

6. PLOS authors have the option to publish the peer review history of their article (what does this mean?). If published, this will include your full peer review and any attached files.

Reviewer #1: No

Reviewer #2: No

---

## [Author Response · Author response to Decision Letter 1]

Dear Editor and Reviewers,

We sincerely thank you for your careful review and valuable comments on our manuscript. We have comprehensively revised the manuscript and provided detailed point-by-point responses to your suggestions. We kindly ask for your continued review and guidance. We are truly grateful for the constructive feedback you provided despite your busy schedules, as it has been greatly helpful in improving the quality of our work.

---

## [Decision Letter · Decision Letter 1]

4 Jun 2025

A Multi-Domain Collaborative Denoising Bearing Fault Diagnosis Model Based on Dynamic Inter-Domain Attention Mechanism and Noise-Aware Loss Function

PONE-D-25-14574R1

Dear Dr. Cao,

We’re pleased to inform you that your manuscript has been judged scientifically suitable for publication and will be formally accepted for publication once it meets all outstanding technical requirements.

Kind regards,

Burak Erkayman

Academic Editor

PLOS ONE

Additional Editor Comments (optional):

Reviewers' comments:

Reviewer's Responses to Questions

**Comments to the Author**

1. If the authors have adequately addressed your comments raised in a previous round of review and you feel that this manuscript is now acceptable for publication, you may indicate that here to bypass the “Comments to the Author” section, enter your conflict of interest statement in the “Confidential to Editor” section, and submit your "Accept" recommendation.

Reviewer #1: (No Response)

2. Is the manuscript technically sound, and do the data support the conclusions?

Reviewer #1: (No Response)

3. Has the statistical analysis been performed appropriately and rigorously? 

Reviewer #1: (No Response)

4. Have the authors made all data underlying the findings in their manuscript fully available?

Reviewer #1: (No Response)

5. Is the manuscript presented in an intelligible fashion and written in standard English?

Reviewer #1: (No Response)

6. Review Comments to the Author

Reviewer #1: This paper has been improved by addressing the comments from reviewers. The quality of this paper has been improved significantly. It can be accepted now.

7. PLOS authors have the option to publish the peer review history of their article (what does this mean?). If published, this will include your full peer review and any attached files.

Reviewer #1: No

---

## [Editor Report · Acceptance letter]

PONE-D-25-14574R1

PLOS ONE

Dear Dr. Cao,

I'm pleased to inform you that your manuscript has been deemed suitable for publication in PLOS ONE. Congratulations! Your manuscript is now being handed over to our production team.

Kind regards,

on behalf of

Dr. Burak Erkayman

Academic Editor

PLOS ONE